# Detecting Bugs with Substantial Monetary Consequences by LLM and Rule-based Reasoning

**Brian Zhang**
University of Texas at Austin
Austin, TX 78705
bz5346@utexas.edu

**Zhuo Zhang**
Purdue University
West Lafayette, IN 47906
zhan3299@purdue.edu

## Abstract

Financial transactions are increasingly being handled by automated programs called *smart contracts*. However, one challenge in the adaptation of smart contracts is the presence of vulnerabilities, which can cause significant monetary loss. In 2024, $247.88 M was lost in 20 smart contract exploits. According to a recent study, accounting bugs (i.e., incorrect implementations of domain-specific financial models) are the most prevalent type of vulnerability, and are one of the most difficult to find, requiring substantial human efforts. While Large Language Models (LLMs) have shown promise in identifying these bugs, they often suffer from lack of generalization of vulnerability types, hallucinations, and problems with representing smart contracts in limited token context space. This paper proposes a hybrid system combining LLMs and rule-based reasoning to detect accounting error vulnerabilities in smart contracts. In particular, it utilizes the understanding capabilities of LLMs to annotate the financial meaning of variables in smart contracts, and employs rule-based reasoning to propagate the information throughout a contract's logic and to validate potential vulnerabilities. To remedy hallucinations, we propose a feedback loop where validation is performed by providing the reasoning trace of vulnerabilities to the LLM for iterative self-reflection. We achieve 75.6% accuracy on the labelling of financial meanings against human annotations. Furthermore, we achieve a recall of 90.5% from running on 23 real-world smart contract projects containing 21 accounting error vulnerabilities. Finally, we apply the automated technique on 8 recent projects, finding 4 known and 2 unknown bugs.

## 1 Introduction

Blockchains are public, append-only ledgers that record data in a secure manner and provide a foundation for many financial applications. Two such applications are tokens and smart contracts. Tokens are cryptocurrencies (e.g., Bitcoin Nakamoto [2008], WETH WETH [2024]). Smart contracts are programs that perform a service once deployed to the blockchain, typically financial in nature. Example services include banking, marketplaces, and loaning. With the digitization of finance, smart contracts have seen increasing usage for financial transactions. For example, the Ethereum blockchain has over 40 million smart contracts deployed Bitkan [2024]. Smart contracts are implemented in a Java-like language known as  Solidity [2023] They are similar to traditional software applications in that they are composed of functions, and not all functions can be invoked (due to function modifiers). *Entry functions* are those that can be directly invoked by *users*, which are defined as any entities (e.g., humans) interacting with the smart contract.

Our work focuses on one challenge in smart contracts, namely their vulnerability to *accounting bugs*. As defined by  Zhang et al. [2023], accounting bugs are incorrect implementations of domain-specific business models. According to the study, accounting bugs are the most popular type of bug, as well as one of the most difficult for humans to identify. Furthermore, they cause significant damage if exploited. In 2024, more than $50 M dollars in damages DefiLlama [2024] were caused by 8

accounting bugs so far. There are many existing vulnerability detection techniques for smart contracts, including fuzzers Wüstholz and Christakis [2020]; Choi et al. [2021] , program analysis Wang et al. [2019]; Huang et al. [2022] , verification Jiao et al. [2020]; Tan et al. [2022] , and symbolic execution Consensys [2024]; Bose et al. [2022]. However, we find that most cannot handle accounting bugs. Recently, Zhang [2024] proposed a program analysis based technique that relies on a type system. In particular, it relies on manual annotation of initial types of variables, which include the currency unit, scaling factor, and correspondence to some functionality in a bank. With such knowledge, it performs type checking to detect accounting bugs. However, the annotation is extensive, in many cases requiring deep understanding of the code and even external documentation. Hence, there is a need to develop an automated technique for detecting accounting bugs in smart contracts.

According to Zhang et al. [2023], the reason that accounting bugs are difficult to identify is due to the necessity to first understand the complex business logic of smart contracts. We hypothesize that LLMs like GPT can be used to perform the analysis, citing their capability to easily comprehend code. For example, Sun et al. [2024] propose an LLM based technique that performs analysis of smart contracts, through utilizing context from past vulnerabilities as well as techniques such as Chain-of-thought (COT) to improve the reasoning capability of the LLM. However, the approach does not focus on accounting bugs, and the analysis is function-level as opposed to file-level. Furthermore, a significant concern with LLM-based techniques is the generation of hallucinations. Hence, a naive approach focused on detecting accounting bugs providing high-level descriptions and few-shot examples of accounting bugs does not suffice, due to the observations that:

**Challenge 1:** *The code of smart contracts provided for analysis is often too large or too costly to fit* in the limited token context space. While it is possible to filter non-relevant functions (i.e., functions which cannot be invoked by the public), the remaining code usually retains the same issues. Furthermore, reducing the scope of the analysis to function-level can miss accounting bugs that spread across multiple functions or even files.

**Challenge 2:** *Most warnings produced by the method are often hallucinations from the model*, and there is no existing technique to validate the warnings without human analysis.

While the naive approach fails, we are able to develop 4 key insights. First, **the task of detecting accounting bugs can be split into two subtasks: assigning financial meanings to variables and checking the correctness of operations**. For example, there is an accounting bug in the operation $Z = X + Y$, where X has a financial meaning of a *balance* (i.e., an amount of some currency) and Y has a financial meaning of *price* (i.e., an exchange rate between two currencies), the reasoning being that such an operation does not result in meaningful output, hence is a violation of all business models. Second, **LLMs such as GPT can easily assign financial meanings to variables through semantic analysis**. However, prompting LLMs to annotate all variables is infeasible due to monetary cost and risks of misclassifications due to hallucinations. Hence our third insight: **rule-based reasoning allows for propagation of financial meaning, efficient checking of operation correctness, and function-level analysis with file-level scope**. Rule-based systems Hayes-Roth [1985] are an approach to artificial intelligence that operates on a set of predefined rules. In our context, inferences rule are utilized for the propagation, taking advantage of the deterministic results of financial meaning operations. Furthermore, the rules substantially reduce the workload on LLMs to only annotating *entry variables* (i.e., parameters of entry functions and global variables), as all other variables can be assigned financial meanings from the propagation. Finally, inference rules can be enhanced to enable propagation through function invocations, reducing the analysis to function-level while maintaining file-level scope for detection. Fourth, while hallucinations are still an issue, **a reasoning trace can be prompted to the LLM to detect hallucinations in annotations**. Furthermore, the trace can be integrated with an iterative feedback loop to automatically remedy hallucinations.

Based on the insights, we develop ABAUDITOR , a hybrid LLM and rule-based reasoning system to automatically detect accounting errors. It utilises the understanding capabilities of LLM (specifically, GPT 3.5-turbo OpenAI [2024]) to provide financial meaning annotations for entry variables and rule-based reasoning to propagate meanings to other variables and to discover potential vulnerabilities. Furthermore, our technique reduces hallucinations by implementing an iterative feedback loop where reasoning traces of potential bugs are provided to the LLM for self-reflection. In sum, we make the following contributions.

- We abstract detecting accounting bugs into two subtasks: identifying financial meanings of variables with LLM, and checking correctness of operations with rule-based reasoning.

```
1  contract MagicLpAggregator{
2  function getPairPrice() public returns (int256) {
3      baseReserve = baseCurrency.totalSupply();
4      quoteReserve = quoteCurrency.totalSupply() * priceOfBaseInQuote();
5      totalShareSupply = totalShareSupply();
6      return ((baseReserve + quoteReserve) / totalShareSupply);
7  }}
```

Figure 1: Buggy Code from *Abracadabra MIMSwap* Abracadabra [2024]

- We develop a method to detect hallucinations in GPT via generating a reasoning trace and prompting it back to GPT for self-reflection.
- We implement a prototype ABAUDITOR based on Slither Feist et al. [2019] using the GPT-3.5 turbo model as our LLM. We evaluate on 34 real contracts from Zhang et al. [2023] that have 40 reported accounting bugs. Among these bugs, we find that 19 of them belong to categories such as *pure math errors* (i.e., the > operation is swapped with <) hence fall out of scope. Our method detects 19 of the remaining in-scope bugs (i.e., 90.5% recall). Furthermore, we achieve 75.6% accuracy in financial meaning annotations by our system versus by humans. We also measure the results without our reasoning trace-based hallucination reduction, and find that the trace reduces the false positive rate by 54.5%. Furthermore, we run our system on 8 new smart contract projects, finding 6 out of 7 accounting bugs, two of which are zero-day (i.e. not discovered before) vulnerabilities.

## 2 Motivation

We demonstrate the effectiveness of our hybrid system with the following real-world example.

In January 2024, the smart contract project *Abracadabra MIMSwap* Mutual [2024] was exploited for $6.5 M dollars. The loss was due to an attack known as a *flash loan attack*, which is when a malicious actor takes advantage of a vulnerability to control a currency's price and make profit. Typically the attack begins via a large loan which is used to skew the price through the vulnerability. The attacker then makes a profit based on the skewed price (by trading the exploited currency), and pays off the loan right after. Flash loan attacks have caused $33.35 M in damages in 2024 alone DefiLlama [2024]. The developers fixed the vulnerability in their new version, and posted the code as a bug bounty on *Code4rena* in March. Code4rena [2024] is a prestigious vendor for smart contract auditing competitions. However, the new implementation was also vulnerable to the aforementioned attack. The vulnerability was an accounting bug, and was only found by two security researchers, among hundreds. Figure 1 shows code for the function that contains the accounting bug. The function is written in Solidity, and has been simplified for demonstrative purposes. The code was taken from the released Code4rena bug report after publication, and the vulnerability has already been fixed.

In Figure 1, getPairPrice() is a function in the file MagicLpAggregator.sol. The file implements an exchange of three cryptocurrencies: the *base token*, the *quote token*, and the *share token*. In the exchange, users can trade base tokens with quote tokens, or vice versa. Intuitively, the functionality is analogous to an ATM. The share token represents a share of the total base and quote token in the exchange. It acts similar to real-world stocks, where instead of a company, ownership is designated to the amounts of base and quote tokens in the exchange. Users can buy or sell the share token by paying or receiving amounts of both the base and quote token, respectively. The function getPairPrice() is used to calculate the price of the share token, in units of the base token.

In the getPairPrice() function, the price of the share token is calculated as follows. The total amount of base tokens is calculated on line 3 via the function call baseCurrency.totalSupply(). The total amount of quote tokens is calculated on line 4 via the function call quoteCurrency.totalSupply(). It is then converted to an equivalent amount of base tokens by multiplying the price of base tokens. The function calls baseCurrency.totalSupply() and quoteCurrency.totalSupply() return the exact amount of each token owned by the smart contract. Then, the amount of available share tokens (totalShareSupply) is obtained on line 5. Finally, the share token price is calculated by adding the total amount of base currency, baseReserve, with the total amount of converted quote currency, quoteReserve, and dividing by the totalShareSupply.

The accounting bug is due to using `baseReserve` and `quoteReserve` to compute the share token price. Specifically, it is due to the function calls `baseCurrency.totalSupply()` and `quoteCurrency.totalSupply()`, which directly return the amounts of base and quote currencies owned by the smart contract. A malicious actor can initiate a flash loan attack by taking a large loan of the base or quote currency and paying it to the smart contract. This results in the amount of base or quote currency to be inflated, and when calling the function `getPairPrice()`, a grossly incorrect share price is returned. The malicious actor can take advantage of the price to make profit.

When our technique is run on the smart contract, it identifies the vulnerability with no false positives. It can determine that the variables `baseReserve`, `quoteReserve`, and `totalPairSupply` all represent reserves in the smart contract, and that the return value of the function `priceOfBaseInQuote()` represents a price. Reserve is defined as an amount of currency that is owned by the smart contract. Price is defined as the exchange rate of one currency to another. Furthermore, it is able to determine that the addition operation on line 6, (`baseReserve + quoteReserve`), also has a financial meaning of reserve. Intuitively, the result of the operation is considered a reserve because it represents the aggregation of two reserve values, thereby maintaining its definition. Our system reports an error for the division operation on line 6, where the code attempts to produce a price by the division of two reserves, following a rule that disallows any division of two reserves. Such divisions are problematic due to being a potential vulnerability for flash loan attacks, as demonstrated above. A better alternative would be to obtain the price by querying *off-chain authorities*, which are trusted services that process data such as prices off the blockchain.

## 3 Design

A naïve approach of directly prompting the LLM does not work ( Appendix A). The under-performance of the naive approach leads to our proposal of ABAUDITOR. Figure 2 shows the architecture of ABAUDITOR. The blue blocks represent actions taken by the system, while the grey blocks

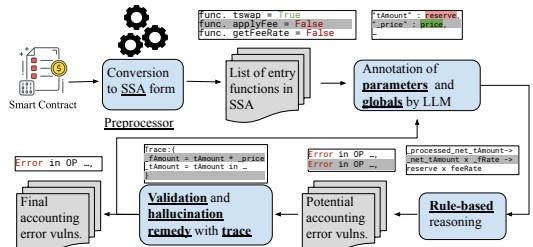

Figure 2: Architecture of Our System

represent the intermediate and final results produced by the system. ABAUDITOR takes as input a singular smart contract file or an entire directory with many such files. First, ABAUDITOR extracts the control flow graph (CFG) of the targeted smart contracts and converts the individual operations to static single assignment (SSA) Cytron et al. [1986], a specific intermediate representation whose details are not needed to understand our paper. From the extracted CFG, *entry functions* are identified. Recall entry functions are those that are accessible to all users. These are done using APIs from an existing analysis framework Slither Feist et al. [2019]. Once all entry functions are identified, the next step involves identifying and annotating the financial meanings of their parameters as well as global variables. This is discussed in more detail in Section 3.1. After the initial labeling, propagation of financial meanings and verification of the correctness of all operations is done through rule-based reasoning. This is discussed in more detail in Section 3.2. At this step, a list of potential vulnerabilities is generated and validated as either a hallucination or a real bug. The validation process is discussed in more detail in Section 3.3

We demonstrate each step of our technique with an example found in Figure 3, which shows a bug. The code contains two functions: `tswap()` and `applyFee()`. `tswap()` is a `public` function (i.e., it can be invoked by all users) which performs a swap of `tAmount` of one currency to another. During the swap, a fee is applied. Function `applyFee()` is an internal function used to calculate the remaining amount after fee is applied in the exchange. The accounting bug is that

```
1  contract CurrencySwap{
2  uint256 netRate;
3  function applyFee(uint256 _tAmount, uint256
       _netRate)
4  internal returns (uint256 _fAmount){
5  _fAmount = _tAmount * _netRate;
6  }
7  function tswap(uint256 tAmount, uint256, price)
8  public returns (uint256 swappedAmount){
9  // USAGE: deducts fee from 'tAmount'
10 fRate = netRate;
11 net_tAmount = applyFee(tAmount, fRate);
12 final_tAmount = net_tAmount * fRate;
13 swappedAmount = final_tAmount * price;
14 }}
```

Figure 3: Demo smart contract with accounting bug

$$+ : (R_1) \ \frac{y_1 : rawbal \quad y_2 : rawbal}{x := y_1 + y_2 \ : rawbal} \quad (R_2) \ \frac{y_1 : reserve \quad y_2 : reserve}{x := y_1 + y_2 \ : reserve}$$

$$- : (R_3) \ \frac{y_1 : rawbal \quad y_2 : rawbal}{x := y_1 - y_2 \ : rawbal} \quad (R_4) \ \frac{y_1 : netbal \quad y_2 : netbal}{x := y_1 - y_2 \ : netbal}$$

$$\times : (R_5) \ \frac{y_1 : rawbal \quad y_2 : netrate}{x := y_1 \times y_2 \ : netbal} \quad (R_6) \ \frac{y_1 : reserve \quad y_2 : netrate}{x := y_1 \times y_2 \ : \textbf{\textcolor{red}{[Error]}}} \quad (R_7) \ \frac{y_1 : netbal \quad y_2 : netrate}{x := y_1 \times y_2 \ : \textbf{\textcolor{red}{[Error]}}}$$

$$\div : (R_8) \ \frac{y_1 : rawbal \quad y_2 : rawbal}{x := y_1 \div y_2 \ : price} \quad (R_9) \ \frac{y_1 : reserve \quad y_2 : reserve}{x := y_1 \div y_2 \ : \textbf{\textcolor{red}{[Error]}}}$$

Figure 4: Subset of Rule-Based Inference Rules

the remaining amount is less than intended. Particularly, this is because fee is applied to it twice, one instance on line 5 in `applyFee()`, and the other on line 12.

## 3.1 Initial Annotation by LLM

For every entry function parameter and global variable, GPT performs initial annotation of their financial meaning. In our implementation, we define 6 possible choices for initial financial meanings, which are: *raw balance*, *net rate*, *interest rate*, *debt*, *price*, and *reserve*. Raw balance is defined as an amount of currency that is strictly owned by a user. Net rate is defined as the percentage left after applying a fee. In the blockchain economy, many smart contracts deduct a fee from users to earn profit. Net rate is the amount left after the fee, (e.g., if 10% fee was deducted, net rate is 90%). Interest rate represents the percentage charged on borrowed or lent funds. Debt refers to the amount of borrowed funds owed by a user. Price represents the exchange rate from one currency to another. Reserve denotes the pool of funds strictly owned by the smart contract. These latter two financial meanings were depicted in Figure 1 in the motivation section. We restrict the choices of initial annotations to these 6 types as they are sufficient for commonly seen business models, although it is very easy to expand the system.

GPT is prompted with high-level definitions of each financial meaning, as well as few-shot examples of real-world instances. To fit the few-shot examples as well as to account for the potential size of the entry function, each financial meaning is formatted in its own prompt. To perform the annotation for an entry function parameter, the name of the parameter and the code of the entry function are provided to the prompt. To perform the annotation for a global variable, a similar prompt is crafted for every function it appears in, and the final assignment is done via the first financial meaning to appear 3 times, or a majority vote. An example can be found in Appendix B and Appendix C.

## 3.2 Rule-based Reasoning

We describe the inference rules used in reasoning. The rules are *invariants* (i.e. properties held across different business models) that we have manually summarized. Th invariants are summarized from a study of many business models and their variants Zhang et al. [2023]. A subset of these rules is included in Figure 4. There are in total 119 such rules. Within the table, $y : \tau$ represents variable $y$ having the financial meaning of $\tau$. The statement $x := ... : \tau$ represents that $\tau$ is the resulting financial meaning from the statement that will be propagated to $x$.

Rule $R_1$ specifies that when a variable $y_1$ with the meaning $rawbal$, or raw balance, is added to a variable $y_2$ which is also a raw balance variable, the result $x$ is a raw balance. Rule $R_2$ is similar for reserve. Rule $R_3$ specifies when a variable $y_1$ is subtracted from a variable $y_2$, and both have financial meanings of raw balance, the result is still a raw balance. Rule $R_4$ specifies when a variable $y_1$ is subtracted from a variable $y_2$, and both have financial meanings of *net balance*, the result is still a net balance. Net balance is defined as an amount of currency owned by a user that has already had a fee applied. Rule $R_5$ specifies when a raw balance variable $y_1$ is multiplied by a net rate $y_2$, the result is a net balance. The operation represents collecting a fee from a user. Rule $R_6$ specifies when a reserve variable $y_1$ is multiplied by a net rate $y_2$, the result is an error. The operation is deducting a fee from the smart contract, which is counterintuitive. Rule $R_7$ specifies when a net balance variable $y_1$ is multiplied by a net rate $y_2$, the result is an error. The operation represents collecting fee twice from a user, which should not be allowed. Rule $R_8$ specifies when a raw balance variable is divided

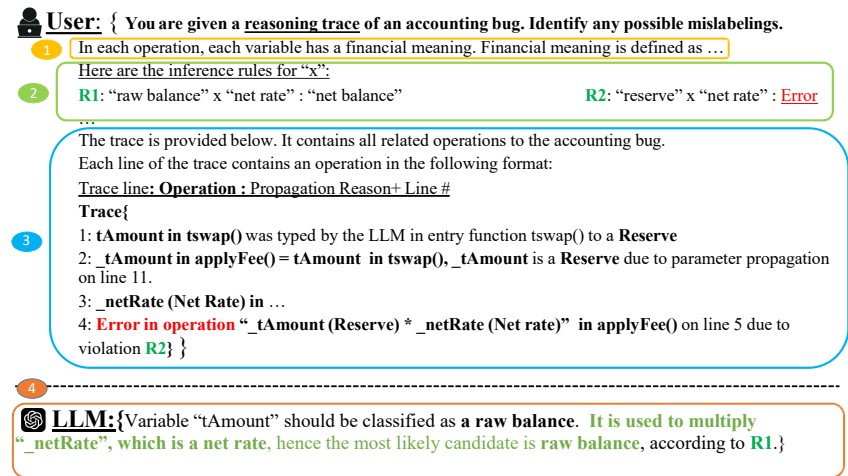

Figure 5: Reasoning Trace Prompt and Response from Hallucination

by a raw balance, the result is a price. Rule $R_9$ specifies when a reserve variable $y_1$ is divided by another reserve variable $y_2$, the result is an error.

The overall process of ABAUDITOR is the following. Given a smart contract, it traverses the statements in the function invocation order and the control flow graph order. It prompts the LLM to assign the initial annotation when encountering a global variable or an entry function parameter that has not been prompted before. Otherwise, it invokes the inference engine that follows the aforementioned rules to propagate financial meanings and identify potential bugs. For each step of propagation, the system records its source. This is for later construction of reasoning traces. For example, _tAmount on line 5 in Figure 3 has its origin from _tAmount on line 3, whose origin is in turn tAmount on line 11, whose origin is line 7 by the LLM. The algorithm and an example of the whole process can be found in Appendix D.

## 3.3 Iterative Validation of Accounting Bugs and Application of Remedy

At this point, our algorithm has produced a list of statements containing potential accounting bugs. However, some of the bugs are produced from mislabelled initial annotations (i.e., due to GPT hallucinations), while others are true accounting bugs. In this subsection, we discuss how we detect hallucinations by providing a reasoning trace for GPT to perform self-reflection. We also explain how this approach can be integrated into an iterative feedback loop to automatically correct hallucinations and re-run the analysis, thereby improving the accuracy of bug detection.

For an accounting bug report regarding some buggy statement, its reasoning trace contains the list of operations involved in the propagation of financial meanings from the initial LLM annotations to the operands in the buggy statement. For each operation, the source statement and the rule applied in propagation are included. At the end, the explanation of why the system considers this a bug is also appended. It provides a comprehensive context on which the LLM performs self-reflection. We then compose a self-reflection prompt to the LLM, providing the trace, the definitions of all financial meanings, together with a subset of related reasoning rules in natural language, and ask the LLM to consider if other initial annotations are possible for the involved entry/global variables. If so, the system continues with the new annotations. If the bug disappears with the new annotations, we consider it a false positive caused by hallucination. The procedure is iterative. If the LLM considers there are no other annotations, and the bug still persists, we consider the bug real and include it in the final report. The trace construction algorithm is presented in Appendix E.

**Example.** An example of a reasoning trace can be found in Figure 5. It was generated from an accounting bug produced by mislabeled inital annotations (due to hallucinations). We have reduced its content for sake of space, and only include necessary details for discussion. Here we discuss the content of the reasoning trace as well as how GPT responds to it. As mentioned previously, the

definitions of all financial meanings are provided in the trace. Specifically, they can be found in the yellow box labeled with "1". The green box labeled with "2" contains the set of relevant rules. We have selected two such rules (**R1**: raw balance × net rate = net balance [a raw balance variable multiplied by a net rate results in a net balance] and **R2**: reserve × net rate = Error). Finally, we provide the list of related operations to the reported accounting bug in the blue box labeled with "3". We go over the included operations as follows. In the first operation, we provide that the variable `tAmount` located in function `tswap()` was initially annotated by GPT as a reserve. In the second operation, we provide that `tAmount` in `tswap()` is propagated to parameter `_tAmount` in function `applyFee()`, and hence the latter also has the financial meaning of reserve. The third line begins a chain of related operations depicting how variable `_netRate` in function `applyFee()` is eventually assigned the financial meaning of net rate, however, we exclude them here for the sake of space. In the fourth operation in the figure, we provide the erroneous operation of `_tAmount * _netRate` on line 5 in `applyFee()`, and explain how it is an accounting bug due to the rule **R2**. In particular, `_tAmount` was assigned the meaning of reserve while `_netRate` was assigned the meaning of net rate, and the result of their multiplication is an error, as defined by **R2**.

The response from GPT can be found in the orange box labelled with "4". In particular, the response details how the entry variable `tAmount` should have been classified as a raw balance. It supports this claim by referencing that if `_tAmount` (which had its meaning directly propagated from `tAmount`) was classified as such, the fourth operation would succeed according to **R1**. Since GPT reports that a reclassification should be made, the initial annotations are updated (i.e., `tAmount` is classified to raw balance), and the rule-based reasoning algorithm is rerun. Although the annotations are now correct, the rerun still produces a potential accounting bug in line 12. We note that this accounting bug is a true bug, as discussed in Section 3.2. The system reports it accurately, and the LLM does not flag any mislabeling. For more details, refer to Appendix F.

# 4 Evaluation

We implement our technique on the Slither [13] framework. We chose GPT-3.5 turbo for our LLM. In this section, we conduct experiments to determine the effectiveness and efficiency of the technique. We aim to answer the following research questions.

- **Research Question 1**: What is the effectiveness of our approach?
- **Research Question 2**: What is the distribution of financial meanings assigned by the LLM? (Appendix G)
- **Research Question 3**: What is the efficiency of the approach?
- **Research Question 4**: What is the effect of fine-tuning and providing few-shot examples in our system?
- **Research Question 5**: Can a less powerful model than GPT-3.5 be used in the approach? (Appendix H)
- **Research Question 6**: How sensitive is the LLM annotation to variable names? (Appendix I)

## 4.1 Experimental Setup

**Benchmark.** We utilize the dataset provided by Zhang et al. [2023], which contains 513 smart contract bugs across 113 smart contract projects. 72 are categorized as accounting bugs. We preclude 15 of them as they cannot be loaded by Slither or miss code. We further exclude 6 projects due to reasons such as code obfuscation. The list of remaining projects can be found in Table 1.While a project may have many files, we only run ABAUDITOR on those with accounting bugs. We further analyze 8 new smart contract projects containing accounting bugs collected from the most recent Code4rena bug reports. The list can be found in Appendix Table 4.

**Baseline.** For our baseline, we ran our rule-based reasoning algorithm using annotations provided by humans on the main dataset. Specifically, we performed manual annotation of financial meanings on every file containing accounting bugs in the benchmark. In each project, we only annotate the entry variables, leveraging the propagation ability of our rule-based reasoning. We find that such effort is substantial, sometimes requiring significant time and attention to detail. We also run our benchmark with two recent tools, namely GPTScan ( Sun et al. [2023]) and ItyFuzz ( Shou et al. [2023]). We do not run with older tools, since according to Zhang et al. [2023] published in 2023, accounting bugs were beyond existing tools at the time.

Table 1: Accuracy of Initial Financial Type Labelling

| Project name | Labelling | | | Baseline | | Ours | | No Rem. | | NIS | Tokens | Func. | Time | Req. |
|---|---|---|---|---|---|---|---|---|---|---|---|---|---|---|
| | Baseline | Ours | Iters. | TP | FP | TP | FP | TP | FP | | | | | |
| MarginSwap | 3 | 3 | 3 | 1 | 0 | 1 | 0 | 1 | 1 | 1 | 37558 | 23 | 24.47s | 18 |
| VaderMath | 4 | 4 | 1 | 2 | 1 | 2 | 1 | 2 | 2 | 2 | 165284 | 60 | 219.74s | 45 |
| PoolTogether | 4 | 2 | 0 | 1 | 0 | 1 | 0 | 1 | 0 | 1 | 51517 | 9 | 10.73s | 16 |
| YieldMicro | 2 | 2 | 1 | 1 | 0 | 1 | 0 | 1 | 2 | 1 | 59592 | 25 | 10.23s | 18 |
| yAxis | 4 | 3 | 1 | 2 | 1 | 2 | 1 | 2 | 2 | 1 | 22178 | 121 | 73.11s | 17 |
| BadgerDao | 4 | 3 | 1 | 1 | 0 | 1 | 0 | 1 | 1 | 0 | 28379 | 22 | 154.06s | 17 |
| WildCredit | 4 | 3 | 0 | 1 | 2 | 1 | 2 | 1 | 2 | 0 | 23058 | 2143 | 87.91s | 14 |
| PoolTogether v4 | 0 | 0 | 0 | 0 | 0 | 0 | 0 | 0 | 0 | 1 | 198600 | 60 | 14.75s | 53 |
| Badger Dao p2 | 5 | 4 | 1 | 1 | 0 | 1 | 0 | 1 | 2 | 0 | 100079 | 4 | 13.71s | 43 |
| yAxis p2 | 1 | 0 | 0 | 0 | 0 | 0 | 0 | 0 | 0 | 1 | 5655 | 5 | 15.2s | 3 |
| Malt Finance | 2 | 0 | 0 | 0 | 0 | 0 | 0 | 0 | 0 | 2 | 139665 | 24 | 51.69s | 45 |
| Perennial | 3 | 3 | 0 | 1 | 0 | 1 | 0 | 1 | 0 | 0 | 4106 | 21 | 146.89s | 7 |
| Sublime | 13 | 9 | 0 | 2 | 2 | 2 | 3 | 2 | 3 | 0 | 27263 | 29 | 35.36s | 18 |
| Yeti Finance | 1 | 1 | 1 | 0 | 0 | 0 | 0 | 0 | 1 | 1 | 14256 | 247 | 48.81s | 14 |
| Vader Protocol p3 | 1 | 1 | 0 | 3 | 0 | 3 | 0 | 3 | 0 | 1 | 90713 | 84 | 121.49s | 47 |
| InsureDao | 6 | 3 | 0 | 0 | 0 | 0 | 1 | 0 | 1 | 1 | 587797 | 32 | 74.06s | 25 |
| Rocket Joe | 4 | 3 | 0 | 1 | 1 | 1 | 1 | 1 | 1 | 0 | 96258 | 41 | 178.76s | 47 |
| Concur Finance | 2 | 1 | 1 | 0 | 0 | 0 | 0 | 0 | 2 | 1 | 62089 | 100 | 184.32s | 40 |
| Biconomy Hyphen | 5 | 4 | 1 | 1 | 0 | 1 | 0 | 0 | 1 | 0 | 105347 | 248 | 102.63s | 30 |
| Sublime p2 | 4 | 4 | 0 | 0 | 0 | 0 | 0 | 0 | 0 | 1 | 109168 | 40 | 34.70s | 37 |
| Volt Finance | 1 | 1 | 0 | 0 | 0 | 0 | 0 | 0 | 0 | 1 | 135919 | 79 | 34.96s | 16 |
| Badger Dao p3 | 4 | 4 | 0 | 0 | 0 | 0 | 0 | 0 | 0 | 1 | 129386 | 104 | 518.63s | 92 |
| Tigris Trade | 5 | 4 | 1 | 1 | 0 | 1 | 1 | 1 | 1 | 1 | 599294 | 3493 | 492.87s | 471 |
| Total | 82 | 62 | | 19 | 7 | 19 | 10 | 19 | 22 | 19 | | | | |

The experiments are conducted on a machine with AMD Ryzen 3975x and 512GB RAM,

## 4.2 Research Question 1: Effectiveness of approach

We ran our system and the baseline technique on the benchmark. We manually inspect all results reported by both methods and categorize them as true positives (TP) or false positives (FP). We also compared the annotations produced by the LLM (i.e., from our system) with the human annotations. We collected the number of invocations of the reasoning trace feedback loop , the number of tokens passed to GPT, the number of total functions analyzed, the run time , and the number of GPT requests. The manual evaluation involved two researchers. Two independently categorized the results and provided annotations while an external judge resolved any inconsistencies.

The results can be found in Table 1. The name of the smart contract project can be found in the leftmost column. The column labelled "Baseline" shows the total human annotations for each project. The column labelled "Ours" shows the number of correct annotations by GPT. The column labelled "Iters" shows the number of times the iterative feedback loop was invoked. The following three headers contain the results from running the baseline (labelled "Baseline"), our technique with the feedback loop (labelled "Ours"), and our technique without it (labelled "No Rem."). For each header, the column labeled with "TP" contains the number of true positive bugs. The column labeled with "FP" contains the number of false positive reports. Accounting bugs that are out of the scope of our system can be found in the column labeled with "NIS" or Not-in-Scope. Such accounting bugs belong to other categories such as pure-math-errors (i.e., a '>' sign is swapped with a '<' sign). The following data are unique to our system. The number of tokens prompted can be found in the column labeled with "Tokens". The number of function analyzed can be found in the column labeled with "Func.". The run time per project can be found in the column labeled with "Time". The number of requests set can be found in the column labeled with "Req.".

**Observations.** Both our system and our baseline were run on 23 smart contract projects, containing 40 accounting errors. In regards to the effectiveness, our system reports 19 TP and 10 FP, while the baseline reports 19 TP and 7 FP. While both techniques are unable to detect 21 of the total accounting errors, we find that 19 of them are beyond scope. Hence, the recalls of both our system and our baseline are 19/(40-19) = 90.5%. In regards to labelling, our technique can correctly annotate

Table 2: Results of GPTScan

| Bug Type | Included Projects | Total Instances |
|---|---|---|
| Wrong Order Interest | 2 | 2 |
| Flashloan Price | 6 | 4 |
| First Deposit | 0 | 0 |
| Approval Not Revoked | 1 | 1 |

Table 3: Results Using Fine-tuned GPTs

| Model | TP | FP | Iters. | Correct Annotations |
|---|---|---|---|---|
| **Baseline** (Manual) | 19 | 7 | N/A | 82/82 |
| **ABAuditor** (Gpt3.5 w/ few-shot) | 19 | 10 | 12 | 62/82 |
| GPT3.5 no few-shot | 17 | 31 | 14 | 32/82 |
| Fine-tuned GPT3.5 no few-shot | 17 | 16 | 9 | 39/82 |
| Fine-tuned GPT3.5 w/ few-shot | 19 | 9 | 7 | 64/82 |
| Fine-tuned GPT4.o mini w/ few-shot | 19 | 9 | 2 | 71/82 |

62 out of the 82 human annotations performed in the baseline, achieving an accuracy of 62/82 = 75.6%. We find that most misclassifications were due to hallucinations in the initial annotation not resulting in accounting bug warnings. As a result, our feedback loop was not run, hence leaving the misclassifications as is. Also observe that using the reasoning traces reduces the number of FPs from 22 to 10, denoting a 54.5% reduction, without losing any TPs. In most cases, fixing these hallucination requires less than 3 iterations.

**False Positives.** We manually inspect the false positives produced by the system. We find that of the 7 FP common to both our full system and our baseline, the most common issue is the path-insensitive nature of our rule-based reasoning. For example, our rules are unable to deal with the scenario where a variable can potentially have two financial meanings due to an `if` conditional. Such scenarios require path-sensitive analysis such as symbolic execution, or enhancing with expressive reasoning methods such as Symbolic Finite Automata (SFA), which can model symbolic transitions between states to model stateful behaviors of business models. Of the 3 FP unique to our full technique, we find that the reason is due to hallucinations in the initial annotation. Particularly, when performing the hallucination remedy procedure, GPT fails to recognize or fails to fix the hallucination. Solving this problem requires improving upon the reasoning trace generation. We will leave resolving both to our future work.

**Further experiments.** To further test the effectiveness of our technique, we run our system on 8 very recent real world smart contracts released in 2024 containing accounting bugs. We do not perform any human annotation, and run our system directly, allowing GPT to perform annotation on all of the entry variables. Our technique can detect 6 of the 7 accounting bugs that are in scope, including 2 zero-day bugs. We have made reports on the zero-day bugs and submitted them to the developers. Our system reports 6 FPs. We argue that this demonstrates the effectiveness of our system even in latest projects. Details can be found in Appendix J.

### 4.3 Research Question 3: What is the efficiency of the technique?

To evaluate the efficiency of the system, we measure the cost of evaluation on the main dataset. Regarding the monetary cost of the system, each project requires an average of 110645 tokens to run. Using the API pricing of the GPT3.5-Turbo model, we find that the average cost to run one such project is $110645 * \$0.5/1000000 = \$0.05$. Regarding the time to run the system, we find that each project requires an average of 115.19 seconds to run, or approximately 2 minutes.

### 4.4 Research Question 4: What is the effect of fine-tuning and providing few-shot examples in our system?

We examined the impact of fine-tuning GPT-3.5 and providing few-shot examples for our system to identify accounting bugs. We also studied the benefits of using a fine-tuned GPT-4.o mini model instead of the GPT-3.5 turbo model. We used 50 fine-tuning examples covering all the supported financial types and those without financial meanings. Then, we evaluated our system with different settings, namely with and without fine-tuning, with and without few-shot examples in prompting..

We include only one setting for GPT-4 due to the high cost of fine-tuning and using it. The results can be found in Table 3. In the table, the first column labelled "Model" shows the settings used for each experiment. The second column labeled "True Pos." shows the true positive bugs detected during each experiment. The column labeled "False Pos." lists the number of false positive warnings reported. The column labeled "Iters." shows the total number of times that the reflection process was run. Finally, the column labeled "Correct Annotations" shows the total number of annotations that match the human-labeled annotations.

Observe that the experiment on row 5 with both fine-tuning and providing few-shot examples to the GPT 3.5 model improved annotation accuracy from 75.6% (62/82; the annotation accuracy of our system located on row 2) to 78% (64/82). Only fine-tuning with no few shots on row 4 performs worse than our default setting. No fine-tuning and no few shots, or simply using the GPT 3.5 model directly, on row 3 generates many more false positives and requires more calls of the reflection process. Using the fine-tuned GPT4.o model has the best performance in terms of a higher annotation accuracy of 86.6% (71/82) at the expense of higher fine-tuning and inference costs. Note that the annotation accuracy changes lead to changes of downstream bug finding. However, the influence may not be proportional because the financial types involved in the bugs are not evenly distributed. That is, the incorrect annotations lie in variables unrelated to the bugs.

## 5 Limitations (see Appendix K)

## 6 Related Work

**Rule-based Systems and Rule-base reasoning.** Rule-based systems are a long-established field in artificial intelligence. Early work such as Hayes-Roth [1985] and Golding and Rosenbloom [1991] derive rule-based systems as a way to modularize knowledge. Rules-based systems have been used to enhance LLMs, such as in generating business insights Vertsel and Rumiantsau [2024] and visual data processing Sharan et al. [2023]. In the context of analysis, rule-based reasoning can be particularly effective, such as in anomaly detection Preece et al. [1992], computer penetration detection Ilgun et al. [1995], and reverse engineering of code Alnusair et al. [2014].

**LLMs in Program Analysis** The use of LLMs in program analysis has gained significant attention in recent years. LLMs leverage extensive datasets and sophisticated neural network architectures to understand and generate human-like text. They have seen usage in tasks such as code understanding Nam et al. [2024], testing Deng et al. [2023], and detecting vulnerabilities Cheshkov et al. [2023]. In relation to our work, LLMs such as GPT have even been used to detect vulnerabilities in smart contracts. Sun et al. [2024] evaluate the vulnerability reasoning capability of LLMs through prompting approaches such as Chain-of-Thought (CoT). Xia et al. [2024] utilize GPT to verify adherence of smart contracts to a set of standards known as the Ethereum Request for Comment (ERC). Sun et al. [2023] perform vulnerability detection via function-level abstraction. In comparison, our system focuses on the detection of accounting bugs in smart contracts.

**Accounting Bugs in Smart Contracts.** A recent technique was proposed in Zhang [2024] to detect accounting bugs using a type system. It relies on intensive manual annotations of initial variable types. A lot of human efforts are hence needed to apply it to a new contract. The technique has inspired our rule-based reasoning part. To some extent, it corresponds to the baseline presented in our evaluation section. In contrast, our technique proposes the novel integration between LLM annotation and rule-based reasoning, enabling automation. It also uses reasoning traces to reduce hallucinations.

## 7 Conclusion

We develop a hybrid LLM and rule-based reasoning system for the detection of accounting bugs in smart contracts. We utilize the understanding capability of LLMs like GPT to perform annotation of financial meanings for variables. We further utilize rule-based reasoning to propagate financial meanings and check for the correctness of operations. We implement a remedy technique for hallucinations, which relies on the generation of reasoning traces. Our results achieve 75.6% accuracy in annotations against those performed by humans, and detects 90.5% of the accounting bugs. Furthermore, our system detects 2 zero-day accounting bugs, which have been reported to the developers.

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

👤 **User**: { **You are given an assistant for identifying accounting bugs in smart contracts.**

> ① Accounting bugs are defined as violations of domain specific business models.
>
> The following are examples of accounting bugs in smart contracts, along with the reason that they are erroneous. Determine if such patterns can be found in the new smart contract.

**Example 1: Code:**{

[Code from Motivating Example]

② `return ((baseReserve + quoteReserve) / totalShareSupply);   (Line 10)`

} **Reason:** { Line 10 is vulnerable to a flashloan attack due to …}

**Example 2: Code:** {…

Please check for accounting bugs in this smart contract:{ [Code for target smart contract] }

③

}

Figure 6: Format for Naive Prompt

👤 **User**: { **You are given an assistant for assigning financial meaning to variables.**

Variables in smart contract often have financial meaning.

> ① The financial meaning "reserve" is any amount of currency that is strictly owned by the smart contract.

The following are examples of "reserve" variables in smart contracts.

**Example 1: Code:**{

`...`

② `quoteReserve = quoteCurrency.totalSupply() * priceOfBaseInQuote();`
`(Line 6)`

`...`} **Variable:** {quoteReserve} **Reason:** {Variable "quoteReserve" is…}

**Example 2: Code:** {…

Answer **YES/NO** if "reserve" is the  financial meaning of **Variable:** {tAmount} in **Function:** {

③ `function tswap(uint256 tAmount, uint256, price) public returns (uint256 swappedAmount){…}    }`

`` ` ``

Figure 7: Format for Initial Prompt for Raw Balance

## Supplementary Material

## A  A Naive Approach

A naive approach is to directly prompt the LLM with few-shot accounting bug examples and ask it to find new bugs of a similar nature. Figure 6 shows the format of the prompt used for the naive approach. Similar to existing works utilizing GPT to detect vulnerabilities in smart contracts Sun et al. [2024]; Shou et al. [2024], high-level descriptions and few-shot examples of accounting bugs are provided within the prompt. Specifically, the blue box labeled "1" contains the definition of accounting bugs, while the red box labeled "2" shows the few-shot examples. Each few shot example follows a two-part format: first, related code is provided, followed by the reason that it contains an accounting bug (i.e., what domain-specific property is violated in the code). The purple box labeled "3" is where the code of a target smart contract is provided as input to the LLM.

However, this approach does not suffice to detect accounting bugs, due to the observations that:

**Challenge 1:** *The code of smart contracts provided for analysis is often too large or too costly to fit* in the limited token context space. While it is possible to filter non-relevant functions (i.e., functions which cannot be invoked by the public), the remaining code usually still retains the same issues. Furthermore, reducing the scope of the analysis to function-level analysis can miss accounting bugs that spread across multiple functions or even files.

**Challenge 2:** *Most warnings produced by the method are often hallucinations from the model*, and there is no existing technique to validate the warnings without human analysis.

## B  Prompt for LLM to Annotate Variable with Finnancial Meanings

Figure 7 demonstrates the prompt format for the financial meaning "reserve". The text in the blue box labelled with "1" contains the definition of a "reserve". The text in the red box labelled with "2" contains few-shot examples for a "reserve" variable. The text in the orange box labelled with "3" contains the prompt used to determine whether or not the new variable (`tAmount`) should be assigned a "reserve".

## C  Example of Initial LLM Annotation

**Example.** Referring to the example provided in Figure 3, there is only one entry function, `tswap()`. `tswap()` contains two parameters: `tAmount` and `price`, which are introduced on line 7. There is also one global variable, `netRate`, which is defined on line 2. Only these 3 variables would be selected for initial annotation, and the correct annotations for the variables are "raw balance", "price", and "net rate", respectively.

## D  Rule-based Reasoning Algorithm and Example

**Data:** $p \leftarrow$ the list of all operations in the smart contract, ordered according to the CFG in SSA
**Result:** $nodes_{error} \leftarrow$
      a list of the operations in SSA form containing accounting error vulnerabilities

1   $nodes_{error} = [\,]$;
2   **while** $p$ is not empty **do**
3      $node_{cur} = p.pop()$;
4      **for** $x$ in $node_{cur}.operands$ **do**
5         **if** ($x$ is entry function parameter or global) and ($x.fin$ is None) **then**
6           $x.fin = $ LLMGetFinanceType($x$);
7         **end**
8      **end**
9      $has_{err}, nodes_{cur}.dest.fin = $ RuleBasedInference($node_{cur}.op, nodes_{cur}.operands$);
10     $node_{cur}.dest.definition = nodes_{cur}$;
11     **if** $has_{err}$ is True **then**
12       $nodes_{error}.add(node_{cur})$;
13     **end**
14 **end**

**Algorithm 1:** Rule-based reasoning pseudocode

We explain the usage of the rule set for analysis and propagation through the pseudocode presented in Algorithm 1. The pseudocode shows the Algorithm for both sections 3.1 and 3.2. $p$ represents the list of operations in the smart contract in CFG order in SSA, and is provided as input. According to Figure 2, it is the first grey box. The output is a list of *nodes* that potentially contain accounting bugs. According to Figure 2, it is the second grey box. A node is defined as either an assignment $x = y$, or a computational operation $x = y_1 \, op \, y_2$. In the pseudocode, $node.dest$ represents the destination of the node (e.g., $x$), $node.op$ represents the operation performed in a node (e.g., $+, -, =, ...$), and $node.operands$ represent the operands (e.g., $y_1, y_2$) that appear in the node. For any variable $x$, its financial meaning is denoted by $x.fin$ and its *parent node* (i.e., the node that assigned its financial meaning) is denoted $x.definition$.

The loop from lines 2-14 traverse $p$ in order, and for each node, performs the following operations. An inner loop from line 4-13 traverses each variable (dubbed $x$) that appears in the node. If variable $x$ is an entry function parameter or a global variable, the initial annotation discussed in 3.1 is performed (lines 5-7). Lines 9-10 show the propagation. The financial meaning of the variable $node_{cur}.dest$, or the destination variable of the current node, is the result of rule-based inference given the operation of the node ($node_{cur}.op$) and its operands ($node_{cur}.operands$). If the operation is valid (i.e., the operation aligns with a rule in the rule set), the destination variable is assigned the resulting financial meaning of the operation. If there is no right operand (i.e., the node is an assignment), the financial meaning of the destination variable is assigned the financial meaning of the singular operand. If the

operation is invalid (i.e., does not satisfy a rule), the node is added to the list of potential accounting bugs, $nodes_{error}$. The definition of variable $node_{cur}.dest$ is assigned to $node_{cur}$.

**Example.** We demonstrate how rule-based reasoning is used to propagate financial meaning as well as identify potential accounting bugs using the example in Figure 3. We use the initial annotations of variable `tAmount` ← reserve, `price` ← price, and `netRate` ← net rate to demonstrate how misclassified initial annotation (i.e., `tAmount` should be a raw balance) can lead to false postive reports.

We begin with the first operation in function `tswap()` of `fRate = getFeeRate()` on line 6. Chaining the SSA nodes of the operation results in: `fRate [in tswap()] = getFeeRate() = feeRate [in getFeeRate()] = feeRate [Global]`. Since all of the SSA nodes are assignments, the financial meaning of `fRate` in `tswap()` is simply that of global variable `feeRate`, which is fee rate (as assigned from the initial annotation.

Continuing to the second operation in function `tswap()` of `net_tAmount = applyFee(tAmount, fRate)` on line 11, we find that there is an improper usage of financial meanings, hence resulting in a potential accounting bug. Specifically, the violation of the rule-based reasoning occurs in the operation `_fAmount = _tAmount * _netRate` on line 5 within function `applyFee()`. In `tswap()`, function `applyFee()` is called with two parameters, `tAmount` (initially annotated as a reserve), and `fRate` (annotated as a net rate). Correspondingly, parameters `_tAmount` and `_netRate` defined on line 4 are assigned the financial meanings of reserve and net rate, respectively. However, the rule-based reasoning is violated on line 5 during the operation `_tAmount * _netRate`. This is because multiplying a reserve and a net rate is not allowed, as demonstrated by rule $R_6$ in Figure 4. Intuitively, the net rate should be applied on a user-owned balance (i.e., raw balance) in order for the smart contract to earn profit, yet this operation shows it being deducting from the smart contract's balance (i.e., reserve), instead. Hence, this node is added to the list of potential accounting bugs.

However, this accounting bug is present due to the misclassification of variable `tAmount` in the initial annotation. By simply replacing the misclassified financial meaning (i.e., reserve) with the correct one (i.e., raw balance), there will be no violation on line 5. Rule-based reasoning applied to the operation `_tAmount * _netRate` yields a valid computation of raw balance multiplied by net rate, as specified by $R_5$ in the rules, leading to the assignment of net balance to the variable `_fAmount`. Additionally, the true accounting bug in the operation `net_tAmount * fRate` on line 12 can be discovered via continuing the analysis. Variable `net_tAmount` on line 11 is assigned a net balance, as it is assigned to be `_fAmount` by the return statement on line 6. In the next operation, `net_tAmount * fRate` on line 11, there is an accounting bug due to the operation net balance * net rate. It is represented by $R_7$ in the rules. Intuitively, the fee has already been deducted from the users balance, as represented by the variable `net_tAmount` being a net balance, hence the operation on line 12 is applying the fee twice, which is not allowed.

## E  Reasoning Trace Construction Algorithm

**Data:** $e$ ← an operation that results in an accounting error vulnerability in SSA
**Result:** $nodes_{trace}$ ← a list of operation in SSA that represent the trace of $e$

1  $nodes_{trace} = []$;
2  $nodes_{wklist} = []$;
3  $nodes_{wklist}.add(e)$ **while** $nodes_{wklist}$ is not empty **do**
4  $\quad node_{cur} = nodes_{wklist}.pop()$;
5  $\quad nodes_{trace}.add(nodes_{cur})$ **for** $x$ in $node_{cur}.operands$ **do**
6  $\quad\quad nodes_{wklist}.add(x.definition)$
7  $\quad$ **end**
8  **end**

**Algorithm 2:** Trace generation pseudocode

To obtain the list of related operations for the response trace, we develop a recursive algorithm that traverses through and tracks each node along with the nodes defining its operands. The pseudocode for the algorithm can be found in Algorithm 2 . The input $e$ is a node that potentially contains an accounting error. The output $nodes_{trace}$ is a list of nodes in the reasoning trace of $e$.

**User:** { You are given a reasoning trace of an accounting bug. Identify any possible mislabelings.

...

① Here are the inference rules for "x":
**R1**: "raw balance" x "net rate" : "net balance"          **R3**: "net balance" x "net rate" : Error

...

② **Trace{**
1: **tAmount in tswap()** was typed by the LLM in entry function tswap() as a **Raw Bal.**
2: **_tAmount in applyFee() = tAmount in tswap(), _tAmount** is a **Raw Bal.** due to parameter prop. on line 11.
3: **_netRate (Net Rate) in** ...
4: **"_fAmount = tAmount * _netRate ", _fAmount** is a **Net Bal.** in **applyFee()** on line 5
        due to rule **R1 (Raw bal. x net rate)**
5: **net_tAmount in tswap() = _fAmount in applyFee(), net_tAmount** is a **Net bal.**
        due to return prop. on line 11.
6: **Error in operation** **"net_tAmount (Net Bal.) * _fRate (Net Rate)" in tswap()** on line 12
        due to violation **R3}**
}

③ ------------------------------------------------------------------------------------------------------------------

**LLM:**{There are no misclassifications. Variable "tAmount" is classified as a raw balance, as it is used in operation with net rate to produce a net balance according to **R1**.}

Figure 8: Reasoning Trace Prompt and Response for Valid Bug

The loop from lines 3-8 contains the procedure applied to a worklist queue that initially contains $e$. First, the current node is popped into $node_{cur}$. The node is added to $nodes_{trace}$. Then, for every variable $x$ in $node_{cur}.operands$, the parent node $x.definition$ is added to the worklist. This process continues until the worklist is empty.

# F  LLM Reflection on a Real Bug Trace

The second reasoning trace in Figure 8 is generated for the accounting bug discovered by the correct initial annotations. While the majority of the content is shared with that in Figure 5 , the green box labeled with "1" representing inference rules contains the rule **R3** net balance × net rate = Error instead of **R2**. In the trace, we note that there are more operations. First, with the relabelling of "reserve" to "raw balance", we find that the fourth operation is a success. The result is that the variable _fAmount in `applyFee()` is now categorized as a net balance. In the fifth operation, _fAmount is propagated to net_tAmount in function `tswap()` on line 11 due to the return statement. In the sixth operation, we provide the new erroneous operation of `net_tAmount * fRate` on line 12, and explain how it is an accounting bug due to the rule **R3**. In particular, `net_tAmount` was assigned the meaning of net balance while `fRate` was assigned the meaning of net rate, and the result of their multiplication is an error, as defined by **R3**.

The response from GPT can be found in the orange box labelled with "4". In particular, the response concludes that no reclassification should be performed. As evidence, it provides the success of operation 4 due to rule **R1**. As such, the accounting bug is considered real.

# G  Research Question 2: What is the distribution of financial meanings?

Figure 9 shows the distribution of the 6 financial meanings in the human annotations performed in the first experiment. Notably, the largest category comprises raw balance variables. This outcome aligns with our expectations. An entry function, defined as one that any user can invoke, often serves as the entrance for users to interact with the smart contract's underlying business model. Hence it is natural that such functions have parameters representing raw balances or amounts of currency owned by users. All other financial meanings are less common in the sense that they are not usually controllable by users.

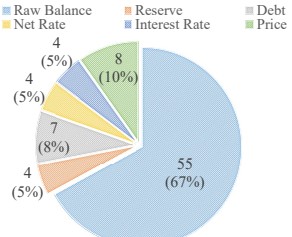

Figure 9: Distributions of financial types

Table 4: Results on New Projects

| Project Name | TP | FP | FN | NIS |
|---|---|---|---|---|
| Abracadabra | 1 | 0 | 0 | 0 |
| Thruster | 0 | 0 | 1 | 0 |
| Acala | 0 | 0 | 0 | 1 |
| Lybra | 1 | 1 | 0 | 0 |
| Polynomial | 2 | 3 | 0 | 0 |
| Trait Forge | 0 | 1 | N/A | N/A |
| Munchables | 1 | 2 | N/A | N/A |
| Basin | 1 | 1 | N/A | N/A |

## H Research Question 5: Can a less powerful model than GPT-3.5 be used in the approach?

We replaced GPT-3.5 in our system with less powerful models such as CodeLlama-7B-instruct-bf and 13B-instruct-hf. CodeLlama-7B failed to produce formatted outputs in many cases such that our pipeline could not parse its results properly. Similar problems were observed in existing works such as Beurer-Kellner et al. [2023]. 13B performs better in this matter. However, it does not seem to understand the nuances of various financial meanings even with the few-shot examples. As such, it produced much worse annotation results than our default setting (41.9% vs 75.6%). This indicates the level of intelligence of the underlying LLM is quite important.

## I Research Question 6: How sensitive is the LLM annotation to variable names?

We conducted an experiment in which we leveraged the Solidity compiler to rewrite variable names (to something as plain as v1, v2), without changing program semantics, to see how the annotation is affected. We kept the function names. We then reran our pipeline with the modified code. Our results showed that the annotation accuracy degraded from 75.6% to 31.7%. The true positives (TPs) degrade from 19 to 14, and the FPs increase from 9 to 21. It is not surprising because without proper variable names, it is extremely difficult even for humans to decide the financial meaning of an operation, e.g., a simple addition. Note that the substantial annotation degradation does not yield in a proportional loss in TPs. This is because for TPs, many incorrect annotations do not happen in the variables that are involved in the bugs. For FPs, the same operations (e.g., simple additions) are allowed for multiple financial types such that even though a variable is mis-annotated, and the system may not flag its operation as an error.

## J Results on Eight New Smart Contracts

The results of running our system on the 8 new smart contracts can be found in Table 4. The name of the project can be found in the leftmost column. The number of true positive accounting errors can be found in the column labeled with "TP". The number of false positive accounting errors can be found in the column labeled with "FP". The number of false negative accounting errors can be found in the column labeled with "FN". The number of accounting errors belonging to other categories (i.e., pure math errors) can be found in the column labeled with "NIS" (Not-in-scope).

Out of the 8 accounting bugs in the new projects, the technique was able to detect 6. Furthermore, one of the undetected accounting errors is due to an incorrect integer value, hence belonging to the pure math category. Of the 6 accounting bugs, 2 are zero-day vulnerabilities, meaning that they were previously undiscovered. Our technique fails to catch 1 bug (FN) and reports 7 false positive bugs (FP). It analyzed a total of 10,097 additional functions. We find that the FN bug is due to the lack of modeling of a specific financial meaning (i.e., a reward rate, which represents the percentage of a loan that is earned as a reward for the loaner). We find that the FP bugs are due to similar reasons, where financial meanings are incorrectly assigned to variables which have financial meanings that are not modeled. We address both in the limitations section of our paper.

## K  Limitations

We note four main limitations: insufficient financial meaning coverage, inability to detect all hallucinations, inability to handle all accounting bugs, and inability to scale to other programming languages. The reason for the first is that we only model 6 financial meanings in our system. However, real-world smart contracts may contain more financial meanings. As such our technique may miss some accounting bugs. This can be improved by expanding the system to accommodate a wider range of financial meanings. The reason for the second is that we only apply the hallucination detection on potential accounting bugs. This can be improved by developing alternative methods to detecting hallucinations. The reason for the third is that our system cannot handle accounting bugs such as pure-math errors. To address this, a more comprehensive approach that combines rule-based reasoning with additional analysis techniques may be necessary to capture these types of errors effectively. For example, our rules could be extended to capture problematic smells, as demonstrated in Rahman et al. [2019]. The reason for the fourth is that Solidity is the most popular programming language for smart contracts, and our implementation currently only supports that. Our system is implemented inside of Slither, a Solidity analysis engine that generates intermediate representations (IR) of smart contracts and provides a rich set of APIs to manipulate the IRS. That said, it is possible to implement the rule checker at the source level (using just a parser), which would allow easy extension to other languages. We leave this to future work.

