# OpenReview forum: "Detecting Bugs with Substantial Monetary Consequences by LLM and Rule-based Reasoning"
_NeurIPS.cc/2024/Conference — NeurIPS 2024 poster_

### Official Review · Reviewer_2yGP · 2024-07-10

**Soundness:** 4
**Presentation:** 4
**Contribution:** 3
**Rating:** 7
**Confidence:** 5

**Summary:**

This study is motivated by accounting bugs in a real-world setting. The authors focused primarily on the Flash Loan attack in smart contracts, which cost $50 million due to eight accounting bugs in DefiLlama.

The proposed method uses a Large Language Model (LLM), GPT3.5-Turbo, to trace a smart contract in the order of the control flow graph (CFG) and generate statements that include possible errors or LLM hallucinations.


A follow-up step involves iterative self-reflection of the GPT via reasoning trace.

**Strengths:**

- Existing vulnerability detection tools can not detect accounting vulnerabilities in smart contracts
- Leverage LLMs to understand the business logic of smart contract
- Analysis of reducing false positives, false positives are a major limitation of static analysis tool
- New bugs were investigated (see Section 4.1).

**Weaknesses:**

- The proposed approach's generalizability is not clearly proved.
 The evaluation of the new smart contact bug is rather minor, with only six true positive bugs.

- The proposed technique uses GPT3.5-Turbo rather than LLM trained on codes like Code Llama[4] or CodeBERT [5].

 **Minor Notes**

-Reducing false positive alarms is a desirable property for security static analysis tools, otherwise, developers will be discouraged from using the tool because of the frequent false positive alarms. The authors could consider the paper [2] to reduce false positive rates of a static analysis tool.

-The authors did not explain how to scale their approach. The authors may potentially scale their technique by taking into account the paper [3].

**Questions:**

1. How were the 119 rules derived? Open coding is a common method for extracting rules from software artifacts, followed by inter-rater agreement. See a sample paper [1].


I am happy to raise my score if the authors address my questions and points mentioned in **Weakness*.*

References:
1. Rahman, A., Parnin, C. and Williams, L., 2019, May. The seven sins: Security smells in infrastructure as code scripts. In 2019 IEEE/ACM 41st International Conference on Software Engineering (ICSE) (pp. 164-175). IEEE.
2. Reis, S., Abreu, R., d'Amorim, M. and Fortunato, D., 2022, October. Leveraging practitioners’ feedback to improve a security linter. In Proceedings of the 37th IEEE/ACM International Conference on Automated Software Engineering (pp. 1-12).
3. Rahman, A., Farhana, E., Parnin, C. and Williams, L., 2020, June. Gang of eight: A defect taxonomy for infrastructure as code scripts. In Proceedings of the ACM/IEEE 42nd International Conference on Software Engineering (pp. 752-764).
4. https://ai.meta.com/blog/code-llama-large-language-model-coding/
5. Feng, Z., Guo, D., Tang, D., Duan, N., Feng, X., Gong, M., Shou, L., Qin, B., Liu, T., Jiang, D. and Zhou, M., 2020, November. CodeBERT: A Pre-Trained Model for Programming and Natural Languages. In Findings of the Association for Computational Linguistics: EMNLP 2020 (pp. 1536-1547).

**Limitations:**

This paper mentions limitations of the proposed approach in Appendix I.

---

> ### Author Rebuttal · Authors · 2024-08-07
>
> ## Response to Reviewer 2yGP
> ### Answers to Questions of Reviewer 2yGP
>
> **Q1. How were the rules derived?**
>
> &nbsp;&nbsp;&nbsp;&nbsp;&nbsp;&nbsp;We are smart contract auditors with years of experience. We have studied many business models (and their variants). These rules are invariants (i.e., properties held across different business models) we manually summarized. We really appreciate the pointer to the inspiring “seven sins” paper [1].  As mentioned in the limitations of our submission (lines 613-615), our rules are not comprehensive. A large empirical study involving more domain experts would help us complete the rule set. In addition, our rules could be extended to capture smells, as according to the paper, smells are quite problematic.
>
> **Generalization to unknown bugs**
>
> &nbsp;&nbsp;&nbsp;&nbsp;&nbsp;&nbsp;Please see C2 of the global response.
>
> **Using CodeLlama and CodeBert**
>
> &nbsp;&nbsp;&nbsp;&nbsp;&nbsp;&nbsp;We tried CodeLlama-7B-instruct-hf and 13B-instruct-hf during rebuttal. CodeLlama-7B failed to produce formatted outputs in many cases such that our pipeline could not parse its results properly. Similar problems were observed in existing works [2,3]. 13B performs better in this matter. However, it does not seem to understand the nuances of various financial meanings even with the few-shot examples. As such, it produced much worse annotation results than our default setting (41.9% vs 75.6%). This indicates the level of intelligence of the underlying LLM is quite important. Please also see C1 in the global response, in which GPT4 has better performance than our default setting (using GPT3.5), namely, 86.6% annotation accuracy (versus 75.6%), fewer FPs, and a much smaller number of iterations before termination (2 vs 12), indicating that it has much fewer hallucinations. However, it also comes with a much higher cost.
>
> &nbsp;&nbsp;&nbsp;&nbsp;&nbsp;&nbsp;CodeBert generates embeddings for both code and natural languages and does not operate like an LLM that can answer our questions. As such, it cannot be used in our pipeline. Exploring integration of code-language models and LLMs is part of our future work.
>
> **Reducing FPs**
>
> &nbsp;&nbsp;&nbsp;&nbsp;&nbsp;&nbsp;Thanks for your pointer to the ASE’22 paper [4]. Although our tool does not produce that many FPs, reducing them is always highly desirable. We will open-source the tool upon publication and plan to solicit feedback from the community through a large-scale study like the one in the paper. The authors have strong connections with both the blockchain industry and the auditing community. These connections will be instrumental for the study.
>
> **How to scale our approach**
>
> &nbsp;&nbsp;&nbsp;&nbsp;&nbsp;&nbsp;Thank you for the pointer to the “Gang of Eight” paper [5]. The scale of the study in the paper and the impact are exemplary. We aim to achieve similar ramifications. Our work mainly followed the taxonomy proposed in [6], which studied 500 or so critical/exploitable real-world smart contract bugs in 2021-2022. We can clearly see the benefits of having a larger study than and a more thorough taxonomy than [6]. From our experience, accounting bugs continue to be a very important bug category, causing substantial monetary loss, and the current tool support is still insufficient. Moreover, we observe that recent smart contracts have increasingly complex business models which demand a more fine-grained study. We plan to conduct a larger scale study focusing on accounting bugs in the past five years and hopefully come up with a fine-grained taxonomy and a comprehensive set of invariants (similar to the rules in our submission). Similar to [1,4,5], we will leverage efforts/feedback from the community.
>
> &nbsp;&nbsp;&nbsp;&nbsp;&nbsp;&nbsp;We will cite [1,4,5] and include the above discussion in the paper.
>
>
> ### References
> > [1] Rahman, A., et al.  “The seven sins: Security smells in infrastructure as code scripts.” ICSE’19.
> >
> > [2] Beurer-Kellner, Luca, et al. "Prompt Sketching for Large Language Models." ICML’24.
> >
> > [3] Beurer-Kellner, Luca, et al. "Prompting is programming: A query language for large language models.", PLDI’23.
> >
> > [4] Reis, S., et al. “Leveraging practitioners’ feedback to improve a security linter.”, ASE’22
> >
> > [5] Rahman, A., et al. “Gang of eight: A defect taxonomy for infrastructure as code scripts.” ICSE’2064).
> >
> >[6]  Zhang, Zhuo, et al. "Demystifying exploitable bugs in smart contracts." ICSE’23.

---

> > ### Author Response · Authors · 2024-08-12
> >
> > Dear Reviewer 2yGP,  thank you very much for your insightful review. We wonder if you have a chance to look into our response. Although we think we have addressed your questions and concerns, we are worried that there may still be places of confusion. We would be very grateful for any feedback you may be able to provide such that we could address your further questions before the interactive period ends.
> >
> > Sincerely,
> >
> > The authors

---

> > ### Author Response · Authors · 2024-08-13
> >
> > Dear Reviewer 2yGP,
> >
> >
> > Thank you for your time and positive review.
> >
> >
> > Since the iterative period is coming to the end, we would be very grateful if you could provide feedback to our response.
> >
> >
> > We believe that we have addressed your question regarding the derivation of rules (question one), generalization to zero-day bugs (weakness one), and using CodeLlma and CodeBert (weakness two), reducing false positives (minor note one), and scaling the technique (minor note two).
> >
> >
> >
> > We would be more than happy to answer any further questions you may have before the period closes.
> >
> >
> >
> > ====================================================================
> >
> > In case you may have further question regarding the quality of the rules, we proactively provide the following explanation.
> >
> >
> > Our rules are comprehensive for common business logics. Note that since these are annotation propagation and checking rules. Problematic rules will lead to failures in type checking. The fact that our system can detect real accounting bugs and rarely has false positives (when typing thousands of functions) demonstrates the quality of our rules. That said, our rules do not model certain rare business behaviors such as options in future contracts. That is why we said our rules are not comprehensive in the limitation discussion of the submission. An analogy is that we are building some type system for Python programs. We design some rules. These rules allow typing common program behaviors, for example, those involving integers and strings. However, we lack rules to type rare behavior of complex numerical values.  The rules are comprehensive and correct for typing the common behaviors, because if they were problematic, many correct Python programs (operating on integers and strings) would be flagged as having type errors, and buggy Python programs would escape the type checking. We will open-source our system and the rules upon publication.
> >
> > Sincerely,
> >
> >
> > The authors

---

> ### Author Response · Authors · 2024-08-09
>
> Thank you for reviewing our paper. We hope that our responses have addressed your concerns. If there are any remaining issues or questions, we would greatly appreciate your feedback so we can make further improvements.

---

> ### Comment · Reviewer_2yGP · 2024-08-13
>
> I would like to thank the authors for their time and addressing my comments.
> I have also read other reviewers' feedback and the authors global rebuttal and I have increased my score.
>
>
> As this paper addresses an important yet underexplored real-world problem, e.g., automatically detecting accounting bugs, I will be happy to see this paper accepted.
>
> I would request the authors to incorporate all reviewers' feedback in the camera-ready version if the paper is accepted.

---

> > ### Author Response · Authors · 2024-08-13
> >
> > Dear Reviewer 2yGP,
> >
> > It seems our messages may have crossed. We greatly appreciate your thoughtful feedback and are pleased that you found our work valuable.
> >
> > We will incorporate all reviewers' feedback in the camera-ready version if the paper is accepted.

---

### Official Review · Reviewer_uXwt · 2024-07-12

**Soundness:** 3
**Presentation:** 3
**Contribution:** 2
**Rating:** 5
**Confidence:** 3

**Summary:**

The work proposes a system to detect accounting bugs in smart contracts by combining large language models (LLMs) and rule-based reasoning. The system first annotates the source code with LLMs to identify the parameters and global variables that are relevant to accounting bugs. Then, the system uses rule-based reasoning to detect accounting bugs by checking the annotated variables against a set of accounting rules. The system is evaluated on 23 real-world projects, and further on 5 recent projects from this year, showing that the system is able to detect accounting bugs effectively.

**Strengths:**

* Ths system combines LLMs and rule-based reasoning to build an effective accounting bug detection system, overcoming the limitations of LLMs.
* The system is able to detect accounting bugs effectively and is further demonstrated by a study of 5 recent projects released this year.
* The technique is fast and cost-effective to run without developers manually annotating the source code, making it potentially useful for a wide range of adoption.

**Weaknesses:**

* The system is not compared with other bug detection systems, and the narrow focus on accounting bugs may limit the applicability of the system. For example, there are about 9 projects out of 23 that are considered "not in the scope" for the system, and even with human annotation, the system cannot detect the bugs. It is not clear how the limitations of the system compare to other bug detection systems.
* The paper would benefit from more analysis on how the LLM is able to perform the annotation. For example, how sensitive is it to the syntax of the code, and is a model less capable than GPT-3.5 enough to do the annotation?

**Questions:**

* It is mentioned that the system only runs on those source code files with accounting bugs; however, in a real-world scenario, the vulnerability is often not known in the first place. What are the false positive rates of the system given a whole project's source code files, assuming the vulnerability location or existence is unknown?
* What portion of the code is included when prompting the model to do annotations of parameters and global variables? Are the relevant code snippets or function implementations included in the prompt?
* How does this work compare with other automatic or semi-automatic bug detection systems? The results table only shows the performance of the proposed system comparing the LLM annotation with human annotation and with/without reflection, and does not compare to other baseline systems.
* How sensitive is the LLM variable annotation to variable names, function names, or comments in the code? Would removing or altering the natural language, such as renaming the variables, affect the annotation accuracy?
* From Table 1, it seems that for some projects, the system didn't annotate correctly for some variables, and still performed the same as the human annotation in terms of TP and FP. Could you provide more insights on why this is the case? What is essential for the system to detect accounting bugs effectively if the annotation is not perfect?
* The reflection process of fixing the annotation may iterate several times. How is it determined when to stop?

**Limitations:**

The limitations are properly addressed in the appendix.

---

> ### Author Rebuttal · Authors · 2024-08-07
>
> ## Response to Reviewer uXwt
> ### Answers to Questions of Reviewer F88y
>
> **Q1. False positive (FP) rate when bugs are unknown**
>
> &nbsp;&nbsp;&nbsp;&nbsp;&nbsp;&nbsp;Please see C2 in the global response.
>
> **Q2. Code snippets included during prompting**
>
> &nbsp;&nbsp;&nbsp;&nbsp;&nbsp;&nbsp;We prompt functions individually. We parse the output of each variable (including global variables). Our rules will propagate such information across functions (e.g., parameter passing from a caller to a callee and uses of the same global variable). In other words, we do not have to prompt a large code body (e.g., including relevant functions) because of the use of rules.
>
> **Q3. Other baselines**
>
> &nbsp;&nbsp;&nbsp;&nbsp;&nbsp;&nbsp;Please see C1 in the global response.
>
> **Q4. Sensitivity to variable names**
>
> &nbsp;&nbsp;&nbsp;&nbsp;&nbsp;&nbsp;We conducted an experiment in which we leveraged the Solidity compiler to rewrite variable names (to something as plain as v1, v2), without changing program semantics. We kept the function names. We then reran our pipeline with the modified code. Our results showed that the annotation accuracy degraded from 75.6% to 31.7%. The true positives (TPs) degrade from 19 to 14, and the FPs increase from 9 to 21.  It is not surprising because without proper variable names, it is extremely difficult even for humans to decide the financial meaning of an operation, e.g., a simple addition. Please also see Q5 to find the explanation of why substantial annotation degradation may not yield proportional loss in TPs.
>
>
> **Q5. Incorrect annotation may not affect TP or FP, and What is essential for the system to detect accounting bugs effectively if the annotation is not perfect?**
>
> &nbsp;&nbsp;&nbsp;&nbsp;&nbsp;&nbsp;For TPs, many incorrect annotations do not happen in the variables that are involved in the bugs. For FPs, the same operations (e.g., simple additions) are allowed for multiple financial types such that even though a variable is mis-annotated, the system may not flag its operation as an error. We will elaborate.
>
> &nbsp;&nbsp;&nbsp;&nbsp;&nbsp;&nbsp;To detect an accounting bug, the annotations for the variables involved in the business operation should be correct.
>
> **Q6. Termination of reflection**
>
> &nbsp;&nbsp;&nbsp;&nbsp;&nbsp;&nbsp;It terminates when the LLM does not propose any new annotations for variables. Note that there are finite types of annotations and hence termination is guaranteed.
>
> **Can a model less powerful than GPT3.5 work?**
>
> &nbsp;&nbsp;&nbsp;&nbsp;&nbsp;&nbsp;We tried CodeLlama-7B-instruct-hf and 13B-instruct-hf during rebuttal. CodeLlama-7B failed to produce formatted outputs in many cases such that our pipeline could not parse its results properly. Similar problems were observed in existing works [1,2]. 13B performs better in this matter. However, it does not seem to understand the nuances of various financial meanings even with the few-shot examples. As such, it produced much worse annotation results than our default setting (41.9% vs 75.6%). This indicates the level of intelligence of the underlying LLM is quite important. Please also see C1 in the global response, in which GPT4 has better performance than our default setting (using GPT3.5), namely, 86.6% annotation accuracy (versus 75.6%), fewer FPs, and a much smaller number of iterations before termination (2 vs 12), indicating that it has much fewer hallucinations. However, it also comes with a much higher cost.
>
> ### References
> > [1] Beurer-Kellner, Luca, et al. "Prompt Sketching for Large Language Models." ICML’24.
> >
> > [2] Beurer-Kellner, Luca, et al. "Prompting is programming: A query language for large language models.", PLDI’23.

---

> > ### Author Response · Authors · 2024-08-12
> >
> > Dear Reviewer uXwt,  thank you very much for your insightful review. We wonder if you have a chance to look into our response. Although we think we have addressed your questions and concerns, we are worried that there may still be places of confusion. We would be very grateful for any feedback you may be able to provide such that we could address your further questions before the interactive period ends.
> >
> > Sincerely,
> >
> > The authors

---

> > > ### Author Response · Authors · 2024-08-13
> > >
> > > Dear Reviewer uXwt,
> > >
> > >
> > > Thank you for your time reviewing our paper.
> > >
> > > Since the iterative period is coming to the end, we would be very grateful if you could provide feedback to our response.
> > >
> > > We believe that we have addressed your five questions regarding false positives and unknown bugs (Q1), code snippets used in prompting (Q2), comparison with baselines (Q3), result sensitivity to variable names (Q4), Table 1 clarification (Q5), and termination of reflection (Q6). We have conducted new experiments and included the results in our response.
> > >
> > >
> > > In addition, we have responded to the two weaknesses: comparison with baselines (w1) and using weaker models (w2).
> > >
> > > Regarding the criticism on narrow focus (part of w1), we have the following further explanation.
> > >
> > > DeFi projects are the most important type of smart contracts. Their over-all market value has reached 103.63B [https://defillama.com/categories]. These projects are susceptible to accounting bugs. Every accounting bug is directly tied to monetary losses. At the time of writing our paper, accounting bugs had caused $50M in damages in 2024, accounting for 25% of the total loss from smart contract exploits Since our paper submission, there was another exploit of an accounting bug that led to the loss of 6.8 million US dollars. Therefore, we consider there is a pressing need to automatically detect such bugs. In addition, real-world accounting bugs are very difficult to detect as they are essentially functional bugs that are contract specific.
> > >
> > > Thank you for your time in advance!
> > >
> > > Sincerely,
> > >
> > > The authors

---

> ### Author Response · Authors · 2024-08-09
>
> Thank you for reviewing our paper. We hope that our responses have addressed your concerns. If there are any remaining issues or questions, we would greatly appreciate your feedback so we can make further improvements.

---

### Official Review · Reviewer_F88y · 2024-07-13

**Soundness:** 2
**Presentation:** 3
**Contribution:** 2
**Rating:** 5
**Confidence:** 4

**Summary:**

The paper introduces ABAUDITOR, a hybrid system that combines LLMs and rule-based reasoning to detect accounting bugs in smart contracts. It leverages the semantic understanding capabilities of LLMs and rule-based logic for validating operations.

**Strengths:**

- The proposed method is computationally inexpensive, facilitating the use of the model on large source code.
- The combination of logical rule-based reasoning with LLMs reduces the model's tendency to produce inaccurate outputs.

**Weaknesses:**

- The primary weakness is that the only baseline used is human annotation; for example, it's not clear how the method compares to those in [1] and [2]. Therefore, the significance of the results remains uncertain.

[1] Zhang, Zhuo, et al. "Demystifying exploitable bugs in smart contracts." 2023 IEEE/ACM 45th International Conference on Software Engineering (ICSE). IEEE, 2023.

[2] Sun, Yuqiang, et al. "Gptscan: Detecting logic vulnerabilities in smart contracts by combining gpt with program analysis." Proceedings of the IEEE/ACM 46th International Conference on Software Engineering. 2024.

**Questions:**

- How many times were the experiments run?
- Please clarify the origin of the 119 rules.
- The manual evaluation process is not sufficiently explained; for example, the number of people involved and how conflicts were handled are not detailed. Please provide more information on these aspects.

**Limitations:**

- In addition to the concerns above, the authors need to justify how scalable the technique is across various programming languages, such as dynamically typed languages like python.
- Please clarify if these 119 rules address all accounting bugs or just a selection of them.

---

> ### Author Rebuttal · Authors · 2024-08-07
>
> ## Response to Reviewer F88y
> ### Answers to Questions of Reviewer F88y
> **Q1. How many times were the experiments run?**
>
> &nbsp;&nbsp;&nbsp;&nbsp;&nbsp;&nbsp;Three due to the cost involved. We will clarify.
>
> **Q2. Origin of the rules**
>
> &nbsp;&nbsp;&nbsp;&nbsp;&nbsp;&nbsp;We are smart contract auditors with years of experience. We have studied many business models (and their variants). These rules are invariants (i.e., properties held across different business models) we manually summarized. We are currently exploring using LLMs to extract a more comprehensive set of such properties. We will elaborate in the paper.
>
> **Q3. Manual evaluation process**
>
> &nbsp;&nbsp;&nbsp;&nbsp;&nbsp;&nbsp;Three people were involved in the manual process. Two were annotating the variables independently, and the third served as a judge when inconsistencies occurred.
>
> **Comparison with [1] and [2]**
>
> &nbsp;&nbsp;&nbsp;&nbsp;&nbsp;&nbsp;We cited [1] and the arxiv version of [2]  in our submission (lines 472-474 and 454-456). [1] is an empirical study of exploitable bugs in real-world  smart contracts and the effectiveness of existing tools on these bugs. It inspired us to work on accounting bugs as the authors found that such bugs are difficult, critical (as they may cause substantial financial loss), and beyond existing tools. [2] used prompt engineering (with Chain-of-Thought) to detect ten bug patterns. Most of them are not related to accounting bugs.
>
> &nbsp;&nbsp;&nbsp;&nbsp;&nbsp;&nbsp;Please also see C1 in the global response.
>
> **Scaling to other programming languages**
>
> &nbsp;&nbsp;&nbsp;&nbsp;&nbsp;&nbsp;Solidity is the most popular programming language for smart contracts, and our implementation currently only supports that. The rule checker is implemented inside Slither, a Solidity analysis engine that generates intermediate representations (IR) of smart contracts and provides a rich set of APIs to manipulate the IRs. That said, it is possible to implement the rule checker at the source level (using just a parser), which would allow easy extension to other languages.
>
> **Comprehensiveness of the 119 rules**
>
> &nbsp;&nbsp;&nbsp;&nbsp;&nbsp;&nbsp;Although these rules cover common business models, as stated in our limitations (lines 612-621), they are not comprehensive. For example, we currently do not model options in future contracts. We leave it for our future work.
>
> ### References
> >[1] Zhang, Zhuo, et al. "Demystifying exploitable bugs in smart contracts." ICSE’23.
> >
> >[2] Sun, Yuqiang, et al. "Gptscan: Detecting logic vulnerabilities in smart contracts by combining gpt with program analysis." ICSE’24.
> >

---

> ### Author Response · Authors · 2024-08-09
>
> Thank you for reviewing our paper. We hope that our responses have addressed your concerns. If there are any remaining issues or questions, we would greatly appreciate your feedback so we can make further improvements.

---

> ### Comment · Reviewer_F88y · 2024-08-12
>
> Based on the responses, the comprehensiveness and clarity of the rules being considered remain unclear. Additionally, the experiments presented in the rebuttal show that ChatGPT-4 significantly outperforms the proposed model, albeit with higher computational costs. This raises concerns about the effectiveness of your model in detecting bugs, especially given its lower performance compared to other techniques, including [1]. Thank you to the authors for their effort in the rebuttal. However, due to concerns about the rules considered and the model's performance relative to other approaches, my score remains unchanged.

---

> ### Author Response · Authors · 2024-08-12
>
> We are grateful for your continuous support and response, which discloses a few places that we did not sufficiently clarify.
>
> 1. **Lower performance compared to [1]**: [1] is an empirical study. It does not propose any technique. One of its important conclusions motivating our work is that accounting bugs are beyond existing automated tools. To confirm this, we double-check their project repo. There were only benchmark programs without any bug detection tools. Our tool is the first fully automated technique that can effectively detect accounting bugs.
>
> 2. **ChatGPT-4**: The reviewer may have some misunderstanding of our GPT-4 results. The reported results are acquired by replacing GPT-3 in our pipeline with GPT-4, not by directly prompting. Our technique is independent of the underlying LLMs. A more powerful LLM yielding better results is indeed a plus for our technique. Directly prompting GPT-4 (with few-shot examples) does not detect accounting bugs. In our experiment, GPT-4 was used to derive the annotations, which were further used in rule-based propagation and checking.
> 3. **Comprehensiveness and clarity of the rules**: our rules are comprehensive for common business logics. Note that since these are annotation propagation and checking rules. Problematic rules will lead to failures in type checking. The fact that our system can detect real accounting bugs and rarely has false positives (when typing thousands of functions) demonstrates the quality of our rules. That said, our rules do not model certain rare business behaviors such as options in future contracts. That is why we said our rules are not comprehensive. An analogy is that we are building some type system for Python programs. We design some rules. These rules allow typing common program behaviors, for example, those involving integers and strings. However, we lack rules to type rare behavior of complex numerical values.  The rules are comprehensive and correct for typing the common behaviors, because if they were problematic, many correct Python programs (operating on integers and strings) would be flagged as having type errors, and buggy Python programs would escape the type checking. We will open-source our system and the rules upon publication.
> 4. **Lower performance than other techniques**: our tool has the state-of-the-art performance in automatically detecting accounting bugs. We have compared with recent automatic tools and tools that require intensive manual annotations. The results show that our tool substantially outperforms in terms of effectiveness and automation. We would be more than happy to empirically compare with any other tools that the reviewer may point us to before the discussion period ends.
>
>
> Thank you again for your time. Please let us know if there are places that we can further clarify.

---

> > ### Author Response · Authors · 2024-08-13
> >
> > Dear Reviewer F88y,
> >
> > Thank you again for your earlier response. We further clarified the concerns mentioned in your response.
> >
> > Since the interactive period is coming to an end, we would be very grateful if you could further comment on our latest response.
> >
> >
> > Sincerely,
> >
> > The authors

---

### Official Review · Reviewer_zmms · 2024-07-27

**Soundness:** 3
**Presentation:** 2
**Contribution:** 2
**Rating:** 5
**Confidence:** 5

**Summary:**

This paper proposes a system to detect accounting error vulnerabilities in smart contracts. The key idea is a hybrid approach that combines LLMs and rule-based reasoning. In particular, it prompts LLMs to annotate the financial meaning of variables in smart contracts, and employs rule-based reasoning to propagate the information throughout a contract’s logic and to validate potential vulnerabilities.  To mitigate hallucinations, the system employs a feedback loop that provides reasoning traces of detected vulnerabilities to the LLM for iterative self-reflection.

The system is applied to 34 smart contract programs written in Solidity which are known to contain 21 accounting bugs out of a total of 40 bugs.  It is able to find 19 of the 21 bugs for a recall of 90.5%. It also achieves 75.6% accuracy in financial meaning annotations compared to humans. The reasoning trace-based hallucination mitigation reduces the false positive rate by 54.5%.

**Strengths:**

1. The overall methodology is very elegant: it combines the strengths of LLMs and logic reasoning, using LLMs to overcome two key weaknesses of rule-based reasoning: annotation inference (pre-reasoning) and trace validation (post-reasoning).

2. The overall empirical results are very strong with both high recall and high precision on a suite of real-world smart contracts.

3. The paper is well-organized and easy to follow (with the exception of the flow being interrupted by frequent references to the appendices).

**Weaknesses:**

Major:

1. It is well known that correctness is critical in the domain of smart contracts; as such, I was expecting a more comprehensive approach in terms of a) coverage of the kinds of bugs detected, and b) expressiveness of the reasoning engine.

The proposed system is very limited in both these aspects: it targets the relatively narrow class of "accounting" bugs -- which cover only 21 out of 40 in the authors' benchmark suite, and the checking is performed using very simple accounting rules.  The system is really a lightweight type qualifier checking engine that is more like a linter than a verifier.

2. There aren't any realistic baselines in the evaluation, namely, existing static analysis and fuzzing techniques.

3. The system uses prompting only as opposed to fine-tuning; the ~ 75% accuracy of annotations indicates that there is significant room to improve, and in any case it would have been nice to see the benefits of fine-tuning.

4. The system has not been applied to find any previously unknown bugs.  The abstract talks about discovering 4 bugs in 5 recent projects, which made me think initially that these were new bugs; but it is later revealed that they were known bugs.

Minor:

- The presentation flow is interrupted by constant references to material in the the appendices.

- Abstract: "Finally, we apply the automated technique on 5 recent projects and discover 4 bugs".  It is a bit misleading to say "discover" since these are known bugs.

- line 3 of Fig 1: Currencey -> Currency

- line 137: totalPairSupply -> totalShareSupply

- I found the Example in Sec 2 to be simplistic; perhaps due to the fact that the approach itself does relatively shallow reasoning.

- line 177: fianncial

**Questions:**

Please see the Major points listed under Weaknesses, especially 2 and 3.

I don't really expect much to change in terms of 1 even though I view it as the single most significant weakness of the work.  However, you could possibly comment on whether your simple rule-based reasoning could be replaced with a more sophisticated reasoning engine, such as an SMT solver.

And while 4 would be nice to address, it is understandable if you did not have the time to apply the system to discover new bugs.

**Limitations:**

Yes, limitations are discussed (although in the Appendix), namely: 1) insufficient financial meaning coverage, 2) inability to detect all hallucinations, and 3) inability to handle all accounting bugs.

However, I don't view these as the most significant limitations (unless by "insufficient financial meaning coverage" the authors are referring to the fact that they target only accounting bugs).  Likewise, 2 and 3 aren't significant since the approach has very high precision and recall.

---

> ### Author Rebuttal · Authors · 2024-08-07
>
> ## Response to Reviewer zmms
> ### Answers to Questions of Reviewer zmms
> **Q1. Replacing with rule-based reasoning with SMT solving and expressiveness of the reasoning technique**
>
> &nbsp;&nbsp;&nbsp;&nbsp;&nbsp;&nbsp;It is possible to enhance our system with an SMT solver. In lines 346-348 of our submission, we mentioned that the most common reason for FPs is the lack of path feasibility reasoning. This can be addressed by using a solver. From our experience, the difficulty of handling accounting bugs may not lie in using inference rules or the much more expressive SMT solving. Instead, it lies in deriving a set of rules/invariants that generalize across a wide spectrum of business models. Complex reasoning rules often run the risk of being too specific to a subset of business models. That said, we do think more expressive reasoning methods such as Symbolic Finite Automata (SFA), which can model symbolic transitions between states could be quite useful in modeling stateful behaviors of business models. It belongs to our future work.
>
> **Q2. Baselines**
>
> &nbsp;&nbsp;&nbsp;&nbsp;&nbsp;&nbsp;Please see C1 in the global response.
>
> **Q3. Benefits of finetuning**
>
> &nbsp;&nbsp;&nbsp;&nbsp;&nbsp;&nbsp;Please see C1 in the global response.
>
> **Q4. Unknown bugs**
>
> &nbsp;&nbsp;&nbsp;&nbsp;&nbsp;&nbsp;Please see C2 in the global response.
>
> **Covering more bug types**
>
> &nbsp;&nbsp;&nbsp;&nbsp;&nbsp;&nbsp;Accounting bugs are quite different from other bugs as they are closely coupled with application specific business logics. Other bugs such as reentrancy and frontrunning have precise definitions and general oracles. To some extent, accounting bugs are analogous to the traditional functional bugs, whereas other smart contract bugs are analogous to buffer-overflow and divided-by-zero bugs. In traditional bug detection, functional bugs are a lot more difficult to deal with than others. In our work, we use type rules, which essentially denote the invariants that all business models should respect, to construct a general detector. Although there may be future designs different from ours, we believe detectors for accounting bugs would likely be different from those for other bug types. Moreover, just like functional bugs, accounting bugs are quite diverse and can be further classified to many sub-categories such as normalization errors, unit errors, interest errors, etc.
>
> **Example in Sec-2 is too simplistic**
>
> &nbsp;&nbsp;&nbsp;&nbsp;&nbsp;&nbsp;In general, accounting bugs are much more complex than that in Sec-2. We chose it for readability. The average reasoning trace for a real bug is 20 lines, where the trace for that example is only 4 lines.
>
> **Minor changes**
>
> &nbsp;&nbsp;&nbsp;&nbsp;&nbsp;&nbsp;Thanks for pointing them out. We will change all of them accordingly.

---

> > ### Author Response · Authors · 2024-08-12
> >
> > Dear Reviewer zmms,  thank you very much for your insightful review. We wonder if you have a chance to look into our response. Although we think we have addressed your questions and concerns, we are worried that there may still be places of confusion. We would be very grateful for any feedback you may be able to provide such that we could address your further questions before the interactive period ends.
> >
> > Sincerely,
> >
> > The authors

---

> > > ### Comment · Reviewer_zmms · 2024-08-12
> > > **Addressed some concerns and score increase**
> > >
> > > I have increased my score in response to the authors' sincere attempt at addressing concerns I raised, including demonstrating that annotation inference can improve with fine-tuning, comparing to more baselines, and finding two previously unknown bugs.
> > >
> > > I am still not convinced about the significance of the contribution since the approach 1) targets a relatively narrow class of bugs, and 2) has lower expressiveness and therefore lower guarantees (than say fuzzing on one extreme and verification on the other).  I still view the proposed approach as a kind of "linter".
> > >
> > > But that said, the approach is novel and the tool seems usable, and so I would be happy with acceptance.

---

> > > > ### Author Response · Authors · 2024-08-13
> > > >
> > > > We are very grateful for your feedback and your support of our paper’s acceptance.
> > > >
> > > >
> > > >
> > > > DeFi projects are the most important type of smart contracts. Their over-all market value has reached 103.63B [https://defillama.com/categories].
> > > >
> > > > At the time of writing our paper, accounting bugs had caused $50M in damages in 2024, accounting for 25% of the total loss from smart contract exploits. Furthermore, accounting bugs are directly tied to monetary losses. Since our paper submission, there was another exploit of an accounting bug that led to the loss of 6.8 million US dollars. Therefore, we consider there is a pressing need to automatically detect such bugs.
> > > >
> > > >
> > > >
> > > > We also very much appreciate your question regarding SMT. It inspired us to revisit our false positive cases. We found that extending our system with dependent types (which utilize SMT solving to achieve path-sensitive analysis to some extent) can eliminate most of them. This will be our immediate future work.

---

> ### Author Response · Authors · 2024-08-09
>
> Thank you for reviewing our paper. We hope that our responses have addressed your concerns. If there are any remaining issues or questions, we would greatly appreciate your feedback so we can make further improvements.

---

### Author Rebuttal · Authors · 2024-08-07

We thank all the reviewers for their time and insightful comments.
## Common Concerns
**C1. Baselines**

&nbsp;&nbsp;&nbsp;&nbsp;&nbsp;&nbsp;According to [1] published in 2023, accounting bugs are beyond existing tools. That is why we did not compare our tool with others. Following the reviewers’ suggestions, we found two recent tools and performed empirical comparison during rebuttal. First, we compared ours with GPTScan [2], a recent LLM-based linter for smart contract bugs. It supports ten common bug patterns, including interest related accounting bugs. The results are shown in Table 1. The first column presents the bug types that it finds at least one instance. The second column lists the number of applications with such bugs, and the third the number of instances found. Only the first row (wrong-order-interest) belongs to accounting bugs, and the two reports in that row are false positives upon inspection. In other words, GPTScan could not find any of the bugs ours found.

**Table 1 GPT-Scan Results**:

| Bug Type      | Included Projects           | Total Instances                |
|-------------|----------------------|---------------------|
| Wrong Order Interest  | 2       | 2    |
| Flashloan Price |  6     | 4         |
| First Deposit  | 3        | 3       |
| Approval Not Revoked | 1 | 1|


We also compared our tool with ItyFuzz [3], a SOTA public smart contract fuzzer. Due to the limited time we have, we ran it for 1-4 hours for each project, corresponding to hundreds of millions of executions per-project. We observed that the coverage reached 32% on average. However, the fuzzer did not report any bugs.

In addition, we compared our tool with fine-tuned GPT3.5 and fine-tuned GPT4 mini. We used 50 fine-tuning examples covering all the supported financial types and those without financial meanings. We then evaluated our system with different settings, namely, with and without fine-tuning, with and without few-shot examples in prompting. The results are shown in Table 2. Due to the high cost of fine-tuning/using GPT4, we only include one setting for it. Observe that finetuned-fewshot (row 5) improved the accuracy of annotations from 75.6%(62/82 in row 2) to 78%(64/82); finetuned-no-fewshot (row 4) performed worse than our default setting (row 2); and no-finetuned-no-fewshot (row 3) has a lot more false positives (31 vs. 10) and more iterations. GPT4 has the best performance with annotation accuracy of 86.6% (71/82). However, its fine-tuning and inference costs are much higher than GPT3.5. Note that the annotation accuracy changes lead to changes of downstream bug finding. However, the influence may not be proportional because the financial types involved in the bugs are not evenly distributed. That is, the incorrect annotations lie in variables unrelated to the bugs.

**Table 2 Results using Finetuned GPTs**

| Model       | True Pos.           | False Pos.  |         Iters.      |  Correct Annotations|
|-------------|----------------------|---------------------|-----------------|------------|
| **Baseline** (_Manual_) | 19| 7| N/A| 82/82|
| **ABAuditor** (_Gpt3.5 w/ few-shot_)  | 19       | 10    | 12| 62/82|
| _Gpt3.5 no few-shot_ |  17  | 31    | 14 |  32/82 |
|  _Fine-tuned GPT3.5 no few-shot_ | 17   | 16  | 9| 39/82  |
| _Fine-tuned Gpt3.5 w/ few-shot_ | 19 | 9 | 7 | 64/82|
| _Fine-tuned Gpt4 mini w/ few-shot_ | 19 | 9 | 2 | 71/82 |


These results support our unique contributions. We will include them in the paper.

**C2. Finding unknown bugs**

&nbsp;&nbsp;&nbsp;&nbsp;&nbsp;&nbsp;During rebuttal, we further scanned 8721 additional functions in the five recent projects used in our submission (lines 355-359). In the submission, we did not scan them as they are not in the files that contain the known bugs. The tool generated 3 reports. However, manual inspection showed that they are not real bugs. On the positive side, our tool does not generate many false warnings in such a large-scale scanning.

In addition, we scanned 75 functions in 3 new projects Munchables [4], Basin[5], and TraitForge[6] from the very recent Code4rena audit competitions whose results are still unknown. We chose them as they are business related. Due to the time constraints, we only scanned the files that define core business logics. We found 2 zero-day (unknown) accounting bugs. The first one  is in function _farmPlots() of file LandManager.sol in project Munchables. It adds a variable of a normalized balance type to another variable of an unnormalized type, which is problematic. The second is in function calcReserve() of file Stable2.sol in project Basin. To validate that they are real, we generated exploits (or POCs) that caused monetary loss. We have submitted the bug reports, which can be found in the one-page supplementary material. In addition to the two bugs, the scanning yielded 4 false positives.

To some extent, these illustrate the level of automation and the effectiveness of our tool.

### References
>[1] Zhang, Zhuo, et al. "Demystifying exploitable bugs in smart contracts." ICSE’23.
>
>[2] Sun, Yuqiang, et al. "Gptscan: Detecting logic vulnerabilities in smart contracts by combining gpt with program analysis." ICSE’24.
>
>[3] Shou, Chaofan, Shangyin Tan, and Koushik Sen. "Ityfuzz: Snapshot-based fuzzer for smart contract." ISSTA’23.
>
>[4] https://code4rena.com/audits/2024-07-munchables#top
>
>[5] https://code4rena.com/audits/2024-07-basin#top
>
>[6] https://code4rena.com/audits/2024-07-traitforge#top

---

### Decision · Program_Chairs · 2024-09-25

**Decision:**

Accept (poster)

**Comment:**

Thanks for your submission to NeurIPS 2024. Overall, the reviewers recognize that the paper presents a novel hybrid approach that combines LLMs and rule-based reasoning to detect accounting error vulnerabilities in smart contracts. The experiment results demonstrate the effectiveness of the proposed method in vulnerability detection in real-world smart contract projects. The reviewers agree that this paper makes important contributions to the community and is valuable to be accepted.